# Knowledge and occupational practices of beauticians and barbers in the transmission of viral hepatitis: A mixed-methods study in Volta Region of Ghana

Silas Adjei-Gyamfi[1,2][☯]*, Abigail Asirifi[3‡], Clotilda Asobuno[4‡], Felix Kwame Korang[5☯]

1 School of Clinical Sciences, Auckland University of Technology, Auckland, New Zealand, 2 Savelugu Municipal Hospital, Ghana Health Service, Savelugu, Northern Region, Ghana, 3 Department of Midwifery, Garden City University College, Kenyasi, Ashanti Region, Ghana, 4 Kassena Nankana West District Health Directorate, Ghana Health Service, Paga, Upper East Region, Ghana, 5 Volta Regional Health Directorate, Ghana Health Service, Ho, Volta Region, Ghana

☯ These authors contributed equally to this work.
‡ AA and CA also contributed equally to this work.
* adjeigyamfis@yahoo.com, sdg2942@autuni.ac.nz

**Data Availability Statement:** The datasets collected, generated, or analyzed during this study have been attached as supplementary information.

## Abstract

### Background

Hepatitis B and C viral (HBV and HCV) infections are endemic in Ghana. Also, the National Policy on Viral Hepatitis stipulates that there is unreliable data, limited knowledge, and a deficiency in research on viral hepatitis, especially among some high-risk workers in the eastern part of the country. This study therefore assessed the knowledge level and occupational practices of street beauticians and barbers in the transmission of HBV and HCV in the Volta Region of Ghana.

### Methods

A cross-sectional mixed methods study was conducted in Volta Ghana from April to June 2021. An in-depth interview was used to collect data from five environmental health officers who were selected as key informants in the qualitative stage. Structured questionnaires/ checklists and direct observations were employed to collect data from 340 street beauticians and barbers in the quantitative stage. During the qualitative stage, the process of coding, and mind mapping via thematic analysis was carried out. Furthermore, descriptive and inferential analyses were performed using Stata version 17.0 at a 95% significance level in the quantitative stage.

### Results

Most street beauticians and barbers reported poor knowledge levels about HBV and HCV (67.0%), although the awareness of this viral hepatitis was high (88.2%). While almost one-third of the participants observed safe occupational practices (31.5%), about 29.0%, 49.4%, and 55.3% of them also followed hand hygiene protocols, wore protective clothes/gloves, and sterilized or disinfected tools after use respectively. Street beauticians and barbers with

**Funding:** This study was funded by The Project for Human Resource Development Scholarship under the Japan International Cooperation Agency (JICA), and Nagasaki University School of Tropical Medicine and Global Health, Japan through MPH scholarship support which was received by FKK from 2019 to 2021. There was no additional external funding received for this study. The funders had no role in study design, data collection and analysis, decision to publish, or preparation of the manuscript.

**Competing interests:** The authors have declared that no competing interests exist.

**Abbreviations:** DEFF, Design effect; EHO, Environmental health officer; EHU, Environmental health unit; HBV, Hepatitis B virus; HCV, Hepatitis C virus; HIV, Human immunodeficiency virus; IPC, Infection prevention and contro; MLGRD, Ministry of Local Government and Rural Development.

higher (tertiary) education (AOR = 6.15; 95%CI = 1.26–29.9; p = 0.024), those who had heavy workload of more than 20 customers per day (AOR = 3.93; 95%CI = 1.26–12.3; p = 0.019), and those who had work experience of at least four years (AOR = 1.65; 95%CI = 1.02–2.69; p = 0.040) were more likely to have good knowledge level about viral hepatitis. Additionally, beauticians were more likely to adhere to safe occupational practices as compared to barbers (AOR = 11.2; 95%CI = 3.46–36.3; p<0.001). The key informant interviews revealed that there was a lack of licensing, monitoring, and planned training for street beauticians and barbers, although their services are rampant in the Volta Region.

## Conclusion

Participants showed high awareness but limited knowledge about HBV and HCV infections. The general safety practices among the participants were poor. Our study results suggest possible viral transmission through the activities of street beauticians and barbers which could be attributed to the lack of regulatory systems and training of these cosmetologists. Policy-makers and regulatory bodies should institute and enforce rigorous policies and guidelines on job-related safety measures and health practices including regular training, monitoring, screening, and vaccination programs for these high-risk community workers in Ghana.

## Introduction

Globally, hepatitis B and C viruses are among other hepatitis viruses (A, D, E) that are of greatest concern due to their high potential for epidemics, disease burden, and mortalities they cause [1, 2]. Approximately more than 350 million and 170 million individuals are infected with hepatitis B virus (HBV) and hepatitis C virus (HCV) respectively with sub-Saharan Africa mostly affected [1]. The prevalence of chronic HBV and HCV infections in Ghana is estimated at 15.6% and 5.4% respectively [2–4]. Additionally, asymptomatic viral hepatitis (HBV and HCV) infections are found to be present in 6.9% and 1.8% respectively of the general adult population in Ho of Volta Ghana [5]. Although HBV and HCV are on the rise, the needed attention has not been given to these viruses as compared to other viral infections like human immunodeficiency virus (HIV) in Ghana. However, the World Health Organization (WHO) has revealed that HBV and HCV are about 50 and 10 times more infectious than HIV correspondingly [1].

Beauticians and barbers work to add prettiness to their customers through shaving and styling hair and trimming nails. Most of these workers from developing countries including Ghana focus more on the decoration and entertainment of their shops than reducing the risks related to their occupation [6]. They render services to most communities in Ghana ignoring that their workplaces and services can be sources of HBV and HCV infections through accidental cuts and abrasions from their occupational tools [7, 8], especially when proper occupational practices and infection prevention and control (IPC) measures are not observed [9, 10]. A study conducted in Nigeria to evaluate preventive methods among commercial barbers found that 10% and 72% of the participants sterilized and disinfected their instruments (hair clippers) respectively while 52% of them used kerosene, an ineffective disinfectant for disinfection procedures. In the same study, the designated brush used for cleaning hair clippers was also used for brushing the customers' hair [8]. In Egypt, Atallah and his colleagues observed that 77% of barbers disinfected used tools, 63% washed their hands, and 62% used protective

equipment when attending to their customers [11]. A similar study in Ghana also reported that most beauticians and barbers diluted their disinfectant (70% alcohol) thereby reducing its sterilization ability. They also observed that the majority of the sterilizer cabinets were not working properly [12].

The ability to prevent viral infections among beauticians and barbers as well as their customers is highly dependent on having adequate knowledge about these infections [6, 12–15]. Some studies discovered that there is an association between knowledge level about HBV and HCV transmission among beauticians or barbers and age, educational status, or working experience [6, 16, 17]. A cross-sectional study in India and Pakistan found that very few barbers (7%) were aware of HBV and HCV infections and that the reuse of blades (razors) is a potential source of HBV and HCV transmission [18, 19]. In Central Ghana, more than half of barbers neither knew how HBV and HCV were transmitted nor believed they could get the infection at their workplaces [6, 20]. In contributing to the 2030 agenda of the World Health Organization (WHO) by eliminating viral hepatitis via the reduction of new infections by 90% and deaths by 65% [1], the occupational activities of street beauticians and barbers and their knowledge level must be of significant concern. Although the transmission of HBV and HCV is publicly understood to happen through unprotected sexual intercourse [2], other practical routes of infection including the operations of beauticians and barbers could be a source of transmission [9, 21]. It calls for studies to assess the knowledge level and occupational practices of beauticians and barbers on viral hepatitis, bring about desirable changes in the behaviour and activities of these cosmetologists, and ultimately reduce the risk of potential transmissions.

The District/Municipal Environmental Health Unit (EHU) in Ghana is a decentralized established body under the Ministry of Local Government and Rural Development (MLGRD), whose services are delivered at the assembly level (metropolitan/municipal/district). Environmental health officers (EHOs) carry out diverse roles and measures for protecting public health, including enforcing environmental health legislation and assessing, and controlling environmental and occupational factors that can potentially affect health. Formerly known as health inspectors, EHOs are responsible for monitoring and enforcing standards of environmental and public health, including food hygiene, work safety, housing pollution control, and preventing environmental health conditions injurious to health [22]. However, as asserted by the Ghanaian Daily Graphic Report in 2013, there is no effective regulatory scheme in Ghana to monitor and guide the activities of some high-risk workers like street beauticians and barbers. Street beauticians and barbers run their private businesses and are only interested in their daily income without considering any safety precautions [23]. Besides the fact that there is unreliable data and deficient occupational safety regulations on viral hepatitis [11, 23], there is also limited research on the awareness of hepatitis and work-related practices among street beauticians and barbers [24] especially in the Volta region of Ghana. This study therefore sought to assess the knowledge level and occupational practices of street beauticians and barbers in the transmission of HBV and HCV infections in the Volta Region of Ghana.

## Methods and materials

### Study area

This study was conducted in the Volta Region of Ghana, which lies within longitudes 00 15'W and 10 15'E, and latitudes 60 15'N and 80 45'N with a surface area of 10,572 km$^2$. It shares borders with the Oti Region in the north, the Gulf of Guinea in the South, the Eastern Region and Lake Volta in the west, and the Republic of Togo in the east. As one of the 16 administrative regions in Ghana, Volta Region has a population of approximately 1,960,000 residing in 2,752 communities which are located in 18 districts or municipalities [25]. The literacy rate in the

region is about 49%. Agriculture is the primary source of income in the region while other technical services like plumbing, barbering, and electrical repairs are gradually on the rise due to increased urbanization [26].

There are 732 health facilities consisting of one teaching hospital, one regional hospital, four polyclinics, 19 district/general hospitals, 215 health centers/clinics, and 482 community-based health planning and services (CHPS) [27]. Tertiary and regional hospitals are responsible for the management of liver conditions, including HBV and HCV infections. The lower-level health facilities largely carry out health education, disease surveillance, referral, and hepatitis B vaccinations. Currently, hepatitis B and C management including laboratory investigations are not covered under the National Health Insurance Scheme (NHIS) and the general population has to make out-of-pocket payments for such services. However, as part of efforts to improve maternal and child health in Ghana via the prevention of mother-to-child transmission of hepatitis B (PMTCT-HepB), all pregnant women are screened across all health facilities as routine perinatal care services. Hepatitis B-positive women are given hepatitis B immunoglobulins with strict adherence to the management protocols. After childbirth, all children are given Hepatitis B vaccine by following the Ghanaian immunization schedule [28].

## Study design

A cross-sectional mixed-methods approach, consisting of a triangulation design was identified to conduct the study involving barbers, beauticians, and EHOs from 29th April to 30th June 2021. The quantitative method was used to provide proven statistical data whereas the qualitative study offered the opportunity to provide meaning behind the numbers. As shown in Fig 1, the quantitative strand was used to assess the knowledge level and occupational practices of street beauticians and barbers. The qualitative strand was employed to conduct key informant interviews with selected EHOs. The study was therefore carried out concurrently thus, a parallel study of both quantitative and qualitative strands within the same study. Triangulation analysis of the survey, observations, and key Informant Interviews was done to form a joint interpretation of the data [29].

## Sample size and sampling

**Quantitative arm.** Applying Cochran 1977 formula with a WHO conservative design effect (DEFF), the sample size (n) was primarily determined as 309 with the following indicators; prevalence (p) of HBV infection in Ghana approximated at 16% [2], margin of error (d) of 5%, standard variate ($Z_{\propto/2}$) at 95% confidence level of 1.96, and WHO conservative DEFF size of 1.5 [30].

Thus, $\dfrac{n = Z_{\propto/2}{}^2 \text{X p}(1-\text{p})}{d^2 \ X \ DEFF = \frac{1.96^2 \ X \ 0.16(1-0.16)}{0.05^2} \ X \ 1.5 = 309}$. The addition of 10% attrition and non-response rate increased the estimated respondents to a final sample size of 340.

In Ghana, the work of street beauticians and barbers is mostly practised in urban areas (capital towns of districts). To recruit the 340 street beauticians and barbers, five districts' administrative capital towns were randomly selected through balloting from the 18 administrative capital towns (used as the sampling unit) in the Volta Region at the initial stage. The selected districts' capital towns included Ho, Aflao, Akatsi, Keta, and Sogakope which are located in Ho, Ketu South, Akatsi South, Keta, and South Tongu districts/municipals respectively. Based on the resident population size of each selected town in 2021, a proportionate sampling technique was employed to sample the respondents as exhibited in Table 1.

Furthermore, due to the difficulty in locating street beauticians and barbers as they temporarily operate from a fixed place and there was no available member list, a snowballing

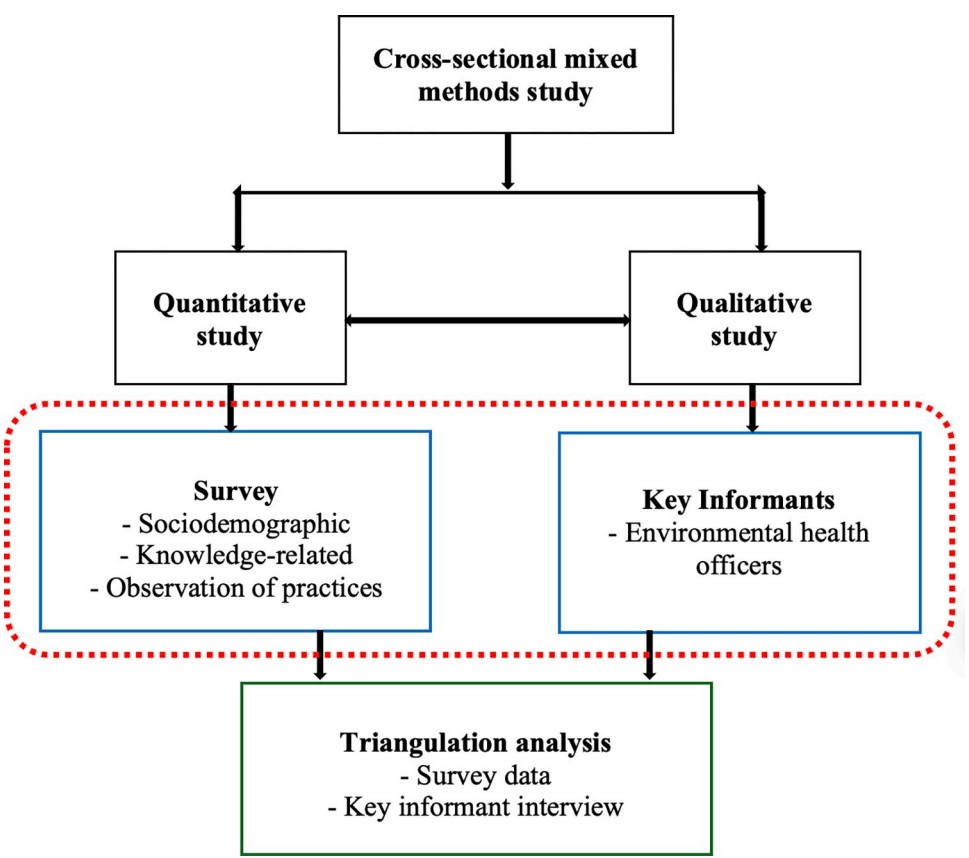

**Fig 1. Mixed methods study design (adapted from Chandi, 2021).**

sampling procedure was used to select the participants until the total sample size of 340 was attained [31, 32]. The first street barber or beautician who met the inclusion criteria was recruited by approaching him/her while providing service to a customer in the community. This technique was also useful as most of the street barbers and beauticians are unregistered and evading tax payments. Therefore, referral by somebody within their social network increased their trust in participating in the study.

## Qualitative arm

Five senior and experienced EHOs were recruited as Key Informants by employing a purposive sampling technique. Thus, one EHO from each of the study sites (district capital town) who also doubles as either the head or deputy manager of the EHU was selected. Targeting one

**Table 1. Description of proportionate sample size for selected towns.**

| Selected towns | Total 2021 population (P) | Town coverage (C = P/718,012) | Estimated sample size (S = C x 340) |
|---|---|---|---|
| Ho | 180,420 | 25.1% | 85 |
| Aflao | 253,122 | 35.3% | 120 |
| Akatsi | 92,494 | 12.9% | 44 |
| Keta | 78,862 | 11.0% | 37 |
| Sogakope | 113,114 | 15.7% | 54 |
| Total | 718,012 | 100% | 340 |

EHO became necessary because each district has at most three EHOs in the district/municipal EHU. The purposive technique was also considered for the selection of these information-rich key informants since they are purported to have an enhanced and diverse understanding of beauticians' and barbers' job practices and associated regulations through long-term service and from classroom training. Notwithstanding, data saturation was used as the guiding principle during recruitment.

## Study population

**Quantitative arm.** The study recruited 340 beauticians and barbers operating in the capital towns of selected districts of the Volta Region. This study did not include beauticians and barbers who were under 18 years old and/or were still undergoing apprenticeship.

**Qualitative arm.** This study targeted five senior and experienced EHOs from the selected district capitals of the Volta Region. However, less experienced officers like junior EHOs who have worked for one or two years were excluded from the study.

**Data collection and variables.** A structured questionnaire, checklist, and interview guide were used to collect information from the study participants. These instruments were designed based on earlier studies [6–8, 11, 19, 20, 33]. The questionnaire had 23 questions that contained information on socio-demographic details, as well as knowledge and awareness about hepatitis B and C. This questionnaire was prevalidated by two other researchers and pretested in Adidome (Central Tongu district) in the Volta Region due to similar characteristics of respondents and was also reliable with a coefficient of Cronbach's alpha of 0.72 [34]. The checklist which had 15 eligibility checks was used to assess the occupational practices of the street beauticians and barbers by ticking on the right work practice from the start of the hair or nail care till completion of the task on a customer. The interview guide was used to assist the data collectors in collecting information on the knowledge and role of the environmental health unit in the operation of street beauticians and barbers in the districts; policy documents, standards, or protocols on practices of beauticians and barbers in Ghana; and regulations, challenges, and recommendations on work practices of street beauticians and barbers.

Seven experienced research assistants (four males and three females) made up of public health officers who are fluent in English and the local dialects (Ewe and Twi) were recruited for data collection. A two-day training session describing an overview of the study, recruitment process, data collection techniques and instruments, consent forms, and confidentiality was carried out for the research assistants. Two research assistants were specifically trained to collect the qualitative data.

## Quantitative arm

A structured questionnaire containing both open-ended and closed-ended questions was used to collect data on beauticians' and barbers' knowledge about HBV and HCV infections and their sociodemographic characteristics. A checklist for direct observation of beauticians and barbers operating on a customer was also carried out. Informed consents were obtained from the beauticians and barbers before the observation. To reduce social desirability bias, the research assistants were selected from outside the study town which put the participants at ease and comfort. The research assistants adopted several approaches to establish friendly rapport with participants and unequivocally explained the details of the study (including the purpose of the study, data use, and confidentiality) to these participants which gave them a better understanding by not seeing the observation as policing or fault finding of their procedures. Likewise, the practices of street beauticians and barbers were observed on two occasions while attending to clients using a checklist without necessarily interacting with them. If the

**Table 2. Description of key variables.**

| Variable | Type of variable | Definition of variable | Categories |
|---|---|---|---|
| Knowledge on viral hepatitis virus | Binary | Estimation of composited knowledge scores using a median cut-off point of 14 knowledge-related items before categorizing into good and poor knowledge level. A knowledge score of < 7 was regarded as 'poor knowledge' while a score of ≥ 7 was regarded as 'good knowledge'. | Good or Poor |
| Occupational practices | Binary | Estimation of composited occupational practice scores using a median cut-off point of 15 practice-related items before categorizing into safe and unsafe occupational practices. A practice score of < 7.5 was regarded as 'unsafe practice' while a score of ≥ 7.5 was regarded as 'safe practice'. | Safe or Unsafe |
| Age group | Ordinal | The age category (in years) of beauticians and barbers. | <20, 20–29, 30–39, ≥40 |
| Gender | Binary | The socially constructed characteristics of participants. | Male or Female |
| Marital status | Nominal | Having customarily or legally bounded partner including being divorced, widowed or cohabiting. | Single, Married, Divorced, Cohabiting |
| Education level | Ordinal | The highest educational level attained by participants. | None, Primary, Junior high, Senior high, Tertiary |
| Occupation | Binary | The type of job for livelihood by participants. | Beautician, Barber |
| Work experience | Binary | The number of working years by participants. | 1–3, ≥4 |
| Daily workload | Ordinal | The average number of customers attended to by participants per day. | <10, 10–19, ≥20 |
| Residence | Binary | The place of residence of participants categorised into urban and rural. | Urban, Rural |

observation could not take place, another date was arranged with the participant. Observations usually took place between 2:00 pm and 6:00 pm on weekdays (Mondays to Fridays) and between 1:00 pm and 6:00 pm on Saturdays and Sundays. These were the prime times the activities of the respondents could be witnessed. If multiple sessions were occurring at the same time, only one session was observed (the first session that began in the presence of the observer).

The outcome variables were knowledge level and occupational practices whereas exposure variables included sociodemographic characteristics. Basic socio-demographic variables including age, gender, marital status, level of education, work experience, and average number of customers per day were also collected. Occupational practices including IPC adopted by each study participant including mode of disinfection of tools, hand washing practices, use of protective clothing, frequent use of a comb, frequency of disinfection, and management of accidental cuts were observed and collected. Participants' knowledge level and understanding of HBV and HCV mode of transmission, signs and symptoms, and preventive measures were explored. The key variables were described and drawn based on existing literature [6, 8, 11, 20] as exhibited in Table 2.

### Qualitative arm

Face-to-face in-depth interviews were conducted for one senior EHO (key informant) in each study site using a semi-structured interview guide to solicit detailed information about street beauticians' and barbers' practices as well as occupational regulations. Upon obtaining the respondents' consent, an iPhone voice recorder was used to record the information while field notes were taken during the interview. Each interview lasted for at least 30 minutes which excluded the time for preparatory and administrative processes.

### Statistical analysis

**Quantitative arm.** STATA version 17.0 (Stata Corporation, Texas, USA) was used to perform all analyses at a significance of p < 0.05. While knowledge, IPC, and occupational-related

categorical variables were reported as frequencies and percentages, continuous variables such as age, work experience, and daily workload were presented as means and standard deviations.

Both knowledge and practice scores were calculated for each participant. Participants received one point for each correct response to the knowledge-related questions and for each correct observed occupational practice. Incorrect responses and observations did not receive any points. A composite knowledge score was calculated using 14 items and dichotomized into good and poor knowledge levels. The practice score was also determined by using 15 itemized practices and categorized into safe and unsafe occupational practices. The reliability of knowledge and practice scores were ascertained and confirmed with the use of coefficients of Cronbach's alpha of 0.81 and 0.79 respectively. By applying the median cut-off point, absolute knowledge and practice scores were estimated since the data were not normally distributed [6, 11, 35–38].

Bivariate associations between knowledge level or occupational practices and sociodemographic variables were assessed using chi-square tests and were confirmed with univariate logistic regression. Significant sociodemographic (predictor) variables at the bivariate level of analysis were forwarded into a hierarchical multivariate logistic regression analysis after addressing multicollinearity issues. Standard logistic regression models were also run. After running a linear regression model to address multicollinearity issues, the predictor variables with a variance of inflation of less than 5 were entered into the multivariate analyses [35, 39]. Meanwhile, biased estimates were attained due to the rareness of the outcome variables and the presence of sparse data for some cells of the variables. Among variables like tertiary education level and occupation, large odds ratios with widened confidence levels were attained. To address this issue, firthlogit binary logistic regression analyses were performed [34, 40, 41]. The "firthlogit" analysis produces virtually unbiased effect size estimates that are representative of the population by maximization of the penalized log-likelihood procedure [34]. Four models were fitted, and Model I served as the null (empty) model. The outcome variable and the key explanatory variables were contained in Model II. Model III contained the outcome variable and socio-demographic characteristics while Model IV contained the outcome variable, the key explanatory variable, and the socio-demographic variables of the participants. The fixed effects parameters were reported as adjusted odds ratios (AORs) at a significance level of $\alpha = 0.05$. The "firthfit" function was used to assess the model fitness comparison by employing penalized log-likelihood, McFadden $R^2$, Akaike's information criterion (AIC), and Bayesian information criterion (BIC). The significant model with the lowest AIC or BIC and/or highest penalized log-likelihood and McFadden $R^2$ values was regarded as the best-fit model to determine the predictors of knowledge level or safety occupational practice among street beauticians and barbers.

**Qualitative arm.** The qualitative data were managed manually, and the analysis was flexible, iterative, and simultaneously conducted during data collection. Pseudonyms were used to store participant information. The audio recordings were repeatedly listened to and transcribed by a team of experts. All field notes were read and compared with the transcribed audio version and then typed into Microsoft Word as the raw data for analysis after verification and review by the authors. The first and last authors revised the transcribed data and read the interview transcripts and field notes several times, to be conversant with the data collected, attain ideas for coding, and identify the main working themes via mind maps. All the authors reviewed the identified themes. A thematic analysis developed by Braun and Clarke (2006) was adopted to identify, analyze, and report patterns within the data collected [42]. Inductive coding was done by employing descriptive coding. This approach gave a comprehensive picture of the themes that emerged from the data. The patterns of similarities and differences were identified from the codes or categories generated, which were put into themes. The codes under

each theme were examined and discussed to ensure that they were descriptive of the themes. Finally, six coding themes were identified to qualitatively assess the occupational and safety practices among street beauticians and barbers. The themes identified were examined, renamed, and triangulated with the quantitative findings to reflect the purpose of the study. Some quotations from participants were extracted as evidence of response.

**Ethical considerations.** Nagasaki University School of Tropical Medicine and Global Health Institutional Review Board (approval number: NU_TMGH_2020_130_1), and the Ghana Health Service Ethics Review Committee (approval number: GHS-ERC 036/01/21) provided ethical clearance for the study. Volta Regional Health Directorate, District/Municipal Health Directorates, and District/Municipal Assembly granted permission for this study. Written informed consent was obtained from all respondents after they were given information on the purpose of the study and what was required of them as study participants.

## Results

### Sociodemographic characteristics among respondents

Table 3 describes the social and demographic variables among street beauticians and barbers. Majority of the 340 beauticians and barbers were between the ages of 20 and 29 years (61.8%) with a mean (sd) age of 26.73 (6.04) years. Although female participants (50.6%) were more than their male counterparts (49.4%), a greater proportion of the street beauticians and barbers were single (61.8%) and had attained junior high educational status (55.9%). Less than half of these beauticians and barbers had worked for at least four years (46.8%), while less than one-third of them resided in rural communities (17.9%) and offered services to approximately 10–15 customers per day (25.0%).

On the other hand, the sociodemographic characteristics of the EHOs are summarised in Table 4. About 80.0% of the EHOs were males and were more than 30 years old. More than half of the officers (60.0%) held the position of deputy district/municipal director and had working experience between 11 to 20 years.

### Knowledge level of viral hepatitis among beauticians and barbers

As illustrated in Table 5, majority of the beauticians and barbers had heard of HBV and HCV before (88.2%). Participants reported that hepatitis B and C can be transmitted through several means; 37.0% and 29.0% mentioned that it can be transmitted through unprotected sex and blood transfusion respectively. Additionally, 20% did not know how HBV and HCV can be transmitted. Almost one-third of the participants mentioned jaundice (28.0%) and muscle/joint pain (27.7%) as symptoms of HBV/HCV infections. Most of the participants did not know that HBV and HCV infections can be transmitted through sharing of their instruments (53.0%) whilst 46.7% did not believe that healthy healthy-looking persons can spread HBV or HCV. Only a few participants perceived themselves to be at risk of contracting HBV or HCV (37.1%). Hence, approximately one-third of these beauticians and barbers had a good knowledge level about HBV or HCV infections (32.9%).

An aspect of the qualitative component of this study highlighted some significant themes underpinning the comprehension of knowledge levels of street beauticians and barbers. As reported by the EHOs, the beauticians or barbers have not received any form of training or education despite the EHOs' greater awareness of their presence and work activities in the entire district. This is a convincing link to why a larger segment of these beauticians and barbers had poor knowledge about viral hepatitis (67.1%). Details of the themes are shown below.

Table 3. Sociodemographic characteristics among beauticians and barbers (n = 340).

| Variables | Frequency distribution | | Knowledge level | | | Occupational practices | | |
|---|---|---|---|---|---|---|---|---|
| | Frequency | Percentage | Good | Poor | χ2 (df), p-value | Safe | Unsafe | χ2 (df), p-value |
| | (n) | (%) | n (%) | n (%) | | n (%) | n (%) | |
| **Age group (years)** | | | | | 0.96 (3), 0.811 | | | 8.20 (3), 0.042* |
| < 20 | 32 | 9.4 | 10 (8.9) | 22 (9.7) | | 10 (9.4) | 22 (9.4) | |
| 20–29 | 210 | 61.8 | 71 (63.4) | 139 (60.9) | | 77 (72.0) | 133 (57.1) | |
| 30–39 | 84 | 24.7 | 28 (25.0) | 56 (24.6) | | 17 (15.9) | 67 (28.8) | |
| 40+ | 14 | 4.1 | 3 (2.7) | 11 (4.8) | | 3 (2.8) | 11 (4.7) | |
| Mean(sd) = 26.73(6.04) | | | | | | | | |
| **Gender** | | | | | 8.11 (1), 0.004* | | | 2.45 (1), 0.049* |
| Male | 168 | 49.4 | 43 (38.4) | 125 (54.8) | | 50 (46.7) | 118 (50.6) | |
| Female | 172 | 50.6 | 69 (61.6) | 103 (45.2) | | 57 (53.3) | 115 (49.4) | |
| **Marital status** | | | | | 5.40 (3), 0.145 | | | 2.17 (3), 0.539 |
| Single | 210 | 61.8 | 62 (55.3) | 148 (64.9) | | 69 (64.5) | 141 (60.5) | |
| Married | 102 | 30.0 | 42 (37.5) | 60 (26.3) | | 27 (25.2) | 75 (32.2) | |
| Divorced | 10 | 2.9 | 4 (3.6) | 6 (2.6) | | 4 (3.7) | 6 (2.6) | |
| Cohabiting | 18 | 5.3 | 4 (3.6) | 14 (6.2) | | 7 (6.5) | 11 (4.7) | |
| **Highest education level** | | | | | 10.1 (4), 0.038* | | | 30.9 (4), <0.001* |
| None | 25 | 7.3 | 5 (4.4) | 20 (8.8) | | 19 (17.8) | 6 (2.6) | |
| Primary | 55 | 16.2 | 18 (16.1) | 37 (16.2) | | 22 (20.5) | 33 (14.2) | |
| Junior high | 190 | 55.9 | 57 (50.9) | 133 (58.3) | | 52 (48.6) | 138 (59.2) | |
| Senior high | 63 | 18.5 | 27 (24.1) | 36 (15.8) | | 12 (11.2) | 51 (21.9) | |
| Tertiary | 7 | 2.1 | 5 (4.5) | 2 (0.9) | | 2 (1.9) | 5 (2.1) | |
| **Occupation** | | | | | 4.90 (1), 0.027* | | | 19.0 (1), <0.001* |
| Beautician | 199 | 58.5 | 37 (33.0) | 104 (45.6) | | 26 (24.3) | 115 (49.4) | |
| Barber | 141 | 41.5 | 75 (67.0) | 124 (54.4) | | 81 (75.7) | 118 (50.6) | |
| **Work experience (years)** | | | | | 3.98 (1), 0.046* | | | 1.39 (1), 0.238 |
| 1–3 | 181 | 53.2 | 51 (45.5) | 130 (57.0) | | 62 (57.9) | 119 (51.1) | |
| 4+ | 159 | 46.8 | 61 (54.5) | 98 (43.0) | | 45 (42.1) | 114 (48.9) | |
| Mean(sd) = 4.57(3.86) | | | | | | | | |
| **Daily workload** | | | | | 6.50 (2), 0.039* | | | 0.94 (2), 0.624 |
| < 10 customers | 241 | 70.9 | 76 (67.9) | 165 (72.4) | | 75 (70.1) | 116 (71.2) | |
| 10–19 customers | 85 | 25.0 | 27 (24.1) | 58 (25.4) | | 29 (27.1) | 56 (24.1) | |
| 20+ customers | 14 | 4.1 | 9 (8.0) | 5 (2.2) | | 3 (2.8) | 11 (4.7) | |
| Mean(sd) = 7.89(4.04) | | | | | | | | |
| **Place of residence** | | | | | 0.07 (1), 0.785 | | | |
| Urban | 279 | 82.1 | 91 (81.2) | 188 (82.5) | | 91 (85.0) | 188 (80.7) | 0.95 (1), 0.331 |
| Rural | 61 | 17.9 | 21 (18.8) | 40 (17.5) | | 16 (15.0) | 45 (19.3) | |

*p-value < 0.05 χ2 = Chi-square df = degree of freedom

## Core duties of environmental health officers

All the EHOs mentioned similar routine daily activities carried out by the district/municipal EHU. However, none of the EHOs mentioned the regulation of beauticians and barbers as part of their core remit. *". . .We do not specialize in one area but carry out all activities that fall within the remit of the department. First and foremost is waste disposal. . ., fumigation of mosquito breeding sites, routine inspection of homes to check hygiene in homes, inspection of eating and drinking areas, market hygiene. . . Meat inspections in abattoirs, cordoning off for dead*

**Table 4. Sociodemographic characteristics of environmental health officers (n = 5).**

| Variables | Frequency (n) | Percentage (%) |
|---|---|---|
| **Age group (years)** | | |
| 20–29 | 1 | 20.0 |
| 30–39 | 2 | 40.0 |
| 40+ | 2 | 40.0 |
| **Gender** | | |
| Male | 4 | 80.0 |
| Female | 1 | 20.0 |
| **Job title** | | |
| District/municipal director | 2 | 40.0 |
| Deputy district director | 3 | 60.0 |
| **Work experience in current position (years)** | | |
| 3–5 | 2 | 40.0 |
| 6–10 | 3 | 60.0 |
| **Total work experience (years)** | | |
| 3–5 | 1 | 20.0 |
| 6–10 | 1 | 20.0 |
| 11–20 | 3 | 60.0 |

*people like COVID-19 patients, we also conduct hygiene inspections in schools, supervise Zoom-lion activities, and screen for food vendors. We cemetery inspectors bury the deceased, people who have died and have no owners. . ., we recommend building plans for approval and prosecuting environmental health violations. . ." EHO4.*

## Awareness of street beauticians and barbers in the district

All EHOs confirmed the presence and operation of street beauticians and barbers in the districts/municipalities. It was also admitted that not much attention has been paid to them regarding their practices and transmission of infections. One respondent narrated: *". . .We know that there are street barbers and beauticians in the district, but we do not have much information about them because they are not under our jurisdiction. We do not inspect them like I said, we only focus on hygiene and food sanitation. . ." EHO3.* Another added: *". . .I know about them. As part of ensuring good hygiene measures, especially for static beauticians, we inspect their environment, especially in terms of how they dispose of their customers' waste water if it has no impact on the environment. As for the barbers, since I have been in the department, we are not as strict. . . I can only remember us visiting the stationed barbers once or twice to check that they had sterilized their machines. . ." EHO5.* One EHO mentioned that the practices of barbers and beauticians pose a very high risk to the public *". . .What I can say is that the practices of some of them are very risky. They can use the same equipment for many people. Those who work in the salon are somehow better, but those who work on the street are very bad. . ." EHO1.*

## Training and continuing education

Typically, beauticians and barbers do not receive regular continuing education to be up-to-date on occupational hazards. It was evident from the responses to the discussions that no form of training has been organized for them. One of the EHO said, *". . . Ahh well, I don't remember any training organized for the street beauticians and barbers since I started work which is more than five years, . . . we may consider one later for them. . ." EHO3.* Another EHO

**Table 5. Knowledge level of viral hepatitis among beauticians and barbers (n = 340).**

| Variables | Frequency (n) | Percentage (%) |
|---|---|---|
| **Ever heard/aware of HBV or HCV infection** | | |
| Yes | 300 | 88.2 |
| No | 40 | 11.8 |
| **Mode of transmission of HBV or HCV** | | |
| Sexual contact | 111 | 37.0 |
| Blood contact/transfusion | 87 | 29.0 |
| Contact with sweat | 66 | 22.0 |
| Sharing of sharp instruments | 34 | 11.3 |
| Contaminated food | 30 | 10.0 |
| Mother to child transmission | 26 | 8.7 |
| Spiritual | 26 | 8.7 |
| Don't know | 60 | 20.0 |
| **Signs & symptoms of HBV or HCV (n = 300)** | | |
| Jaundice | 84 | 28.0 |
| Swollen abdomen | 51 | 17.0 |
| Fatigue | 79 | 26.3 |
| Muscle and joint pain | 83 | 27.7 |
| Don't know | 3 | 1.0 |
| **HBV or HCV can be transmitted by your instrument (n = 300)** | | |
| Yes | 141 | 47.0 |
| No | 159 | 53.0 |
| **Healthy looking person can spread HBV or HCV (n = 300)** | | |
| Yes | 160 | 53.3 |
| No | 140 | 46.7 |
| **Do you perceive yourself at risk for HBV or HCV** | | |
| Yes | 126 | 37.1 |
| No | 214 | 62.9 |
| **HBV can be prevented by vaccination** | | |
| Yes | 220 | 64.7 |
| No | 120 | 35.3 |
| **Number of doses of vaccine that should ideally be given to prevent HBV** | | |
| One | 97 | 28.5 |
| Two | 190 | 55.9 |
| Three | 53 | 15.6 |
| **Overall knowledge level** | | |
| Good | 112 | 32.9 |
| Poor | 228 | 67.1 |

added: "*. . .. Imagine there has been no form of training for even the static barbers and beauticians for over years,. . . errh I can't recall it . . ., so it means I have never thought of training for the street ones. . .*" *EHO2.*

## Occupational practices among beauticians and barbers

Table 6 shows the daily routine occupational practices including IPC among street beauticians and barbers. Out of the 340 beauticians and barbers observed, none of them had a standard protocol to guide their operations whilst only 10.3% had ever received any form of training on IPC practices. A larger percentage of them used 70% alcohol-based disinfectant for

**Table 6. Occupational practices among beauticians and barbers.**

| Variables | Frequency (n) | Percentage (%) |
|---|---|---|
| **Have standards for operating** | | |
| Yes | 0 | 0 |
| No | 340 | 100 |
| **Ever received training on IPC practices** | | |
| Yes | 35 | 10.3 |
| No | 305 | 89.7 |
| **Type of disinfectant used** | | |
| None | 41 | 12.1 |
| 70% alcohol | 250 | 73.5 |
| Dettol | 2 | 0.6 |
| Hot water | 5 | 1.5 |
| Perfume | 22 | 6.5 |
| Soap solution | 9 | 2.6 |
| Talcum powder | 11 | 3.2 |
| **Dilutes alcohol before use (n = 250)** | | |
| Yes | 140 | 55.2 |
| No | 110 | 44.8 |
| **Frequency of tool disinfection/sterilization** | | |
| Before using on a client | 15 | 4.4 |
| After using on every client | 188 | 55.3 |
| Close of day's work | 137 | 40.3 |
| **Perform hand hygiene before attending to clients** | | |
| Yes | 100 | 29.4 |
| No | 240 | 71.6 |
| **Perform hand hygiene after attending to clients** | | |
| Yes | 73 | 21.5 |
| No | 267 | 78.5 |
| **Had accidental cut** | | |
| Yes | 15 | 4.4 |
| No | 325 | 95.6 |
| **Managing accidental cuts (n = 15)** | | |
| No treatment | 4 | 26.7 |
| Cleaned with cotton wet alcohol | 6 | 40.0 |
| Applied only cotton/foam | 2 | 13.3 |
| Applied herbal ointment | 2 | 13.3 |
| Applied a piece of hair | 1 | 6.7 |
| **Wear protective clothes and gloves** | | |
| Yes | 168 | 49.4 |
| No | 172 | 50.6 |
| **Use the same towel for >1 client** | | |
| Yes | 91 | 26.7 |
| No | 249 | 73.3 |
| **Use the same brush/comb for >1 client** | | |
| Yes | 200 | 58.8 |
| No | 140 | 41.2 |
| **Dispose off waste into wastebins** | | |
| Yes | 276 | 81.1 |

(*Continued*)

**Table 6.** (Continued)

| Variables | Frequency (n) | Percentage (%) |
|---|---|---|
| No | 64 | 18.9 |
| **Change tool/blade for each client** | | |
| Yes | 263 | 77.4 |
| No | 77 | 22.6 |
| **Overall occupational safety practices** | | |
| Safe | 107 | 31.5 |
| Unsafe | 233 | 68.5 |

disinfection procedures (73.5%) and 12.1% did not have any form of disinfectant. More than half of the participants diluted the 70% alcohol-based disinfectant before use (55.2%) and sterilized or disinfected reusable tools after using them on a customer (55.3%). Out of the 15 accidental injuries that were observed, only 40% cleaned the wound with cotton wet with alcohol.

Additionally, during their services with a customer, it was observed that most of the participants used a new blade or tool for every customer (77.4%), and the same hairbrush or comb for more than one customer (58.8%). While only a few used the same towel for more than one customer (26.7%) and practiced hand hygiene before attending to a customer (29.4%), majority of the participants disposed of used materials in a dustbin (81.1%). Furthermore, slightly less than half of the participants did not wear protective clothes (especially gloves) when attending to a customer (49.4%). Surprisingly, nearly one-third of the street beauticians and barbers offered safe occupational services to the citizens of Volta Region (31.5%).

On the other hand, the interview with the EHOs suggests that there are no licensing, regulations, or monitoring exercises for the occupational activities of street beauticians and barbers. This accounts for the non-safety practices among most street beauticians and barbers (68.5%) in the districts. Additionally, the EHOs encounter several challenges in the regulation or monitoring exercises and recommend the formation of an association of street beauticians and barbers in the districts as illustrated in the themes below.

## Licensing and regulation

In terms of licensing and regulation of beauticians and barbers in Ghana, it was found that the various districts/municipalities do not necessarily license or regulate their activities. All the EHOs indicated that there is no policy or regulatory system in place to monitor occupational safety practices among street beauticians and barbers. The responses of the discussants suggest that the various local assemblies are only fixated on the taxes paid by street beauticians and barbers and have little or no concern for their practices. The lack of regulatory mechanisms for the activities of street beauticians and barbers can lead to unhealthy practices that expose the public and themselves to blood-borne infections such as HBV and HCV. The following quotes support the findings: "*. . . No, we do not have any laws or policies regulating barbers. All I know is that barbers or beauticians working in a kiosk have to apply for a business operating permit (BOP) at the Municipal Assembly before they start, but street barbers don't do any registration at all. . . We have laws related to sanitation and food hygiene issues, but we don't have laws for the operation of street barbers. . ."* EHO4. Another EHO said: *". . .We have no standards or laws available to us, we made bylaws to contain all social services, but they have not been gazetted. . . . what we do currently is if a beautician or a barber wants to set up a business, he/she has to come to the district assembly to take BOP."* EHO2.

### Challenges among environmental health officers

One bottleneck to the inability of EHOs to monitor and enforce the safety practices of street barbers and beauticians is the lack of clear environmental regulations and laws/regulations. Participants also cited the influence of local political leaders and insufficient funds to purchase equipment and other expenses such as fuel and logistics as major obstacles to the effective monitoring of beauticians and barbers. These quotes support the views of the EHOs. *". . .We have problems with staff, no vehicles. We only have one motorcycle for our transportation. We fund most of our own transportation for our inspections. Regulations issued to help regulate social services have not yet been gazetted. . ." EHO2.* Another said: *". . .Apart from the fact that there is no law to regulate street barbers, we are also affected by the interference of politicians. You know, if you are a law enforcement officer and a criminal is arrested, before you take the person to prosecution, you get a call from the District Chief Executive or whoever expecting you to let the culprit go. . ." EHO5.*

### Suggestions to initiate regulation of beauticians and barbers

Participants made some suggestions for better regulation of beauticians' and barbers' practices which included the provision of bylaws, policy documents, and the formation of a cosmetology association. *". . .We need something like bylaws to monitor their activities, so we have something to rely on. . ." EHO2. ". . .We should have a clearly defined policy document that regulates the activities of barbers and beauticians. . ." EHO3. ". . .As it is now, we have no idea about their total number and how to track them down because almost all of them are on the move. Regulating the street beauticians and barbers is very difficult in the first place, I think we need to encourage them to form an association. . . ." EHO4. ". . .The Assembly should provide us with the necessary resources and funding. . ." EHO1.*

### Factors associated with knowledge level and occupational practices

After registering no multicollinearity among the five predictor variables (gender, educational level, occupation, work experience, daily workload) that showed significant bivariate association with knowledge level (Table 3), three of them namely educational level, work experience, and daily workload remained statistically significant at the multivariate analysis level (Table 7). Moreover, four variables including age, gender, educational level, and occupation were independently associated with safe occupational practices during bivariate analysis (Table 3). Out of these variables, only occupation status showed a significant association in the final model (Table 8).

   The findings suggest that street beauticians and barbers who had higher education up to the tertiary level were 6.15 times more likely to have good knowledge about HBV and HCV infections (AOR = 6.15; 95%CI = 1.26–29.9; p = 0.024). In addition, participants who had a workload of more than 20 customers per day were about four times more likely to have good knowledge about viral hepatitis (AOR = 3.93; 95%CI = 1.26–12.3; p = 0.019). Beauticians and barbers who have been in the profession for at least four years were 65% more likely to have adequate knowledge about viral hepatitis (AOR = 1.65; 95%CI = 1.02–2.69; p = 0.040). Finally, street beauticians had increased odds of adhering to safety occupational (work) practices compared to street barbers (AOR = 11.2; 95%CI = 3.46–36.3; p<0.001).

## Discussion

Local cosmetologists commonly called beauticians and barbers serve as a major source of income for several young adults in the Volta Region of Ghana. Since there is no entry

**Table 7. Factors associated with knowledge level.**

| Variables | Model I (Null) | Model II | Model III | Model IV |
|---|---|---|---|---|
| | A0R (95%CI) | A0R (95%CI) | A0R (95%CI) | A0R (95%CI) |
| *Fixed effects results* | | | | |
| **Occupational practices** | | | | |
| Unsafe | – | Reference | – | Reference |
| Safe | – | 1.29 (0.79, 2.13) | – | 1.26 (0.73, 2.17) |
| **Gender** | | | | |
| Male | – | – | Reference | Reference |
| Female | – | – | 1.56 (0.55, 4.44) | 1.44 (0.50, 4.15) |
| **Education level** | | | | |
| None | – | – | 0.71 (0.23, 2.21) | 0.74 (0.24, 2.31) |
| Primary | – | – | 1.17 (0.60, 2.29) | 1.19 (0.61, 2.33) |
| Junior high | – | – | Reference | Reference |
| Senior high | – | – | 1.76 (0.96, 3.22) | 1.74 (0.95, 3.18) |
| Tertiary | – | – | 6.23 (1.27, 30.4)* | 6.15 (1.26, 29.9)* |
| **Occupation** | | | | |
| Barber | – | – | Reference | Reference |
| Beautician | – | – | 1.41 (0.48, 4.09) | 1.58 (0.52, 4.74) |
| **Work experience (years)** | | | | |
| 1–3 | – | – | Reference | Reference |
| 4+ | – | – | 1.67 (1.03, 2.71)* | 1.65 (1.02, 2.69)* |
| **Daily workload** | | | | |
| <10 persons | – | – | Reference | Reference |
| 10–19persons | – | – | 1.08 (0.61, 1.93) | 1.10 (0.61, 1.96) |
| 20+ persons | – | – | 3.94 (1.26, 12.3)* | 3.93 (1.26, 12.3)* |
| **Model parameters** | Model I | Model II | Model III | Model IV |
| Constant | Reference | 0.41 (0.27, 0.62)*** | 0.20 (0.11, 0.34)*** | 0.16 (0.08, 0.34)*** |
| Number of observations | 340 | 340 | 340 | 340 |
| Wald $\chi 2$ | Reference | $\chi 2\,(1) = 1.07$ | $\chi 2\,(9) = 24.9$ | $\chi 2\,(10) = 25.5$ |
| Prob > $\chi 2$ | Reference | 0.301 | 0.003 | 0.004 |
| Penalized log likelihood | −188.9 | −211.4 | −190.5 | −188.9 |
| **Model fit statistics** | | | | |
| AIC | Reference | 428.8 | 411.1 | 410.8 |
| BIC | Reference | 440.3 | 468.5 | 466.9 |
| McFadden $R^2$ | Reference | 0.003 | 0.069 | 0.071 |

*$p < 0.05$

**$p < 0.01$

***$p < 0.001$

AOR = Adjusted odds ratio; CI = Confidence interval; AIC = Akaike's information criterion; BIC = Bayesian information criterion

impediment to working as a street beautician or barber in the country, this mixed methods study was aimed to assess the knowledge and occupational practices of street beauticians and barbers in the transmission of viral hepatitis (HBV and HCV) in the Volta Region of Ghana.

## Knowledge about HBV and HCV among beauticians and barbers

Our study reported a higher level of awareness about HBV and HCV infections than the level reported among barbers and beauticians in Kumasi City, Ghana [6], although most of this

**Table 8. Factors associated with occupational practices.**

| Variables | Model I (Null) | Model II | Model III | Model IV |
|---|---|---|---|---|
| | A0R (95%CI) | A0R (95%CI) | A0R (95%CI) | A0R (95%CI) |
| *Fixed effects results* | | | | |
| **Knowledge level** | | | | |
| Poor | – | Reference | – | Reference |
| Good | – | 1.30 (0.79, 2.13) | – | 1.29 (0.75, 2.21) |
| **Age group (years)** | | | | |
| < 20 | – | – | Reference | Reference |
| 20–29 | – | – | 1.11 (0.48, 2.58) | 1.11 (0.48, 2.57) |
| 30–39 | – | – | 0.62 (0.24, 1.59) | 0.62 (0.24, 1.59) |
| 40+ | – | – | 0.63 (0.15, 2.71) | 0.61 (0.14, 2.63) |
| **Gender** | | | | |
| Male | – | – | Reference | Reference |
| Female | – | – | 0.19 (0.06, 0.61)** | 0.28 (0.08, 1.59) |
| **Education level** | | | | |
| None | – | – | 2.96 (0.98, 8.93) | 2.92 (0.96, 8.83) |
| Primary | – | – | 1.33 (0.67, 2.61) | 1.34 (0.68, 2.63) |
| Junior high | – | – | Reference | Reference |
| Senior high | – | – | 0.68 (0.34, 1.39) | 0.71 (0.35, 1.45) |
| Tertiary | – | – | 1.07 (0.22, 5.13) | 1.18 (0.24, 5.78) |
| **Occupation** | | | | |
| Barber | – | – | Reference | Reference |
| Beautician | – | – | 11.3 (3.48, 36.6)*** | 11.2 (3.46, 36.3)*** |
| **Model parameters** | **Model I** | **Model II** | **Model III** | **Model IV** |
| Constant | Reference | 0.38 (0.26, 0.58)*** | 0.25 (0.11, 0.60)** | 0.21 (0.08, 0.54)** |
| Number of observations | 340 | 340 | 340 | 340 |
| Wald χ2 | Reference | χ2 (1) = 1.07 | χ2 (9) = 38.3 | χ2 (10) = 38.6 |
| Prob > χ2 | Reference | 0.301 | <0.001 | <0.001 |
| Penalized log likelihood | −188.9 | −207.7 | −173.3 | −171.6 |
| **Model fit statistics** | | | | |
| AIC | Reference | 421.3 | 374.7 | 370.3 |
| BIC | Reference | 432.8 | 428.3 | 426.5 |
| McFadden R$^2$ | Reference | 0.003 | 0.140 | 0.143 |

*$p < 0.05$

**$p < 0.01$

***$p < 0.001$

AOR = Adjusted odds ratio; CI = Confidence interval; AIC = Akaike's information criterion; BIC = Bayesian information criterion

study's participants had poor knowledge levels (67.1%). This suggests a gradual increase in awareness of HBV and HCV infections among Ghanaian cosmetologists. The interview of the key informants also disclosed that there had been no training and/or continuing education for the street beauticians and barbers, although the EHOs are aware of the practices of these local cosmetologists in the Volta Region. This finding substantiates the report by Ghana Daily Graphic [23] and adds to the reason why a majority of the street beauticians and barbers had poor knowledge levels about viral hepatitis.

A significant proportion of beauticians and barbers reported wrong information about HBV and HCV transmission through contaminated food and spiritual means and also could

not tell how viral hepatitis can be transmitted. Most of the participants did not know that HBV and HCV can be transmitted by sharing instruments and did not consider themselves at risk of contracting HBV and HCV whereas less than half of them did not believe that a healthy-looking person can transmit HBV and HCV. These results from the present study are parallel to other studies in sub-Saharan Africa including Ghana which also reported that most barbers or beauticians neither know how HBV and HCV are transmitted nor consider themselves at risk of getting these infections through their work activities [6, 20] while other studies were contradictory [21, 43]. It indicates that, although the majority of street beauticians and barbers were aware of HBV and HCV infections, increased misconceptions are circulating among the Ghanaian populace. Hence, as suggested by the Ghana Daily Graphic Report [23] and the EHOs in the qualitative component of this study, there should be provision of policy and/or bylaws for the active training, regulation, and licensing of these street beauticians and barbers in Ghana.

## Occupational practices among beauticians and barbers

The observed practices of the participants showed that none of the street beauticians and barbers had a protocol for IPC and management of accidental cuts. The study findings do not only conform with a cross-sectional study in the middle belt of Ghana [6] but are also dissimilar to some studies in Egypt and Turkey [11, 44]. This study also revealed that only a few of the participants had received training on IPC practices which gives credence to the comments made by EHOs that street beauticians and barbers had not ever received training from the EHU in the districts/municipals. It implies that there is a lack of oversight responsibility of the practices of street beauticians and barbers by the EHU and relevant authorities as reported by the EHOs. As Ghana continues to face a high burden of viral hepatitis infections [2], conscious effort needs to be devoted to IPC and safety work mechanisms among these local cosmetologists. In this regard, increasing the knowledge levels of beauticians and barbers and modifying their practices are required to reduce occupational viral transmission. Additionally, they could be trained to serve as viral disease (HBV, and HCV) peer educators due to their unique access to the general population [19, 44]. Even though the majority of beauticians and barbers seemed to observe some form of disinfection procedures, most of them were not performed adequately and efficiently. Majority of the participants used 70% alcohol as a disinfectant while more than half of them had diluted it. This finding is consistent with a similar study in Southern Ghana, which reported that most barbers diluted the 70% alcohol reducing its sterilizing ability [12] but incongruent with a study in Nigeria [8]. Several studies have shown that an alcohol concentration of less than 60% cannot inactivate HBV and HCV [12, 45]. Other street beauticians and barbers from our study used inappropriate disinfectants such as herbal ointment, soap solution, and perfume which are not known to act as disinfectants against HBV and HCV. In line with other study results [8, 46], our study participants may use these unacceptable disinfectants for economic purposes and therefore need continuing education on IPC measures to comprehend the health benefits of appropriate disinfectants. These practices are an infringement of professional social responsibility to protect customers [45, 47] and therefore validate that the seemingly high rate of disinfection activities by beauticians and barbers only gives customers and the public a false sense of safety.

The study results showed that 27% and 59% of participants used the same towel and comb or hair brush on all customers without disinfecting it, respectively. Almost half of the participants used protective clothes but admitted to reusing them on other clients. In terms of frequency of disinfection, a substantial number of beauticians and barbers disinfected their tools after providing service to a customer. Further observed occupational safety practices showed

that less than one-third of street beauticians and barbers washed their hands before and after attending to a client. Majority of them disposed of used blades and other sharps directly into dustbins instead of recommended sharps disposal containers. However, they changed the blade used on their previous customers. These findings are not different from results in other parts of Africa [6, 12, 48, 49] but are contrasting to some surveys across Europe and Asia [13, 21, 38, 43, 50]. Micro-traumatic injuries caused as a result of beauticians' and barbers' procedures on their customers can contaminate their tools including combs and hair brushes, and the re-use of these tools can result in the plausible spread of HBV and HCV via micro-abrasions [19]. Pathogens can also be transferred from infected sources to used gloves and aprons, making them an alternative potential source or carrier of viral transmission [10, 21, 45]. It is therefore recommended for cosmetologists to frequently sanitize and disinfect used tools every time between every customer during hair or nail care [45]. This practice becomes imperial as HBV can survive on surfaces for seven days or longer and can infect any susceptible person exposed to it during contact times [1]. Furthermore, the majority of the participants who did not wash their hands could pick up pathogens like HBV or HCV from a customer unknowingly and possibly transmit the virus to him/herself or the next customer. The practice of improper waste disposal by most cosmetologists has continuously been posing a risk to the environment mostly to children and people who search for recycling items at the refuse dumps in Ghana [27] which could also serve as a medium of viral transmission. Nonetheless, due to the gradual increase in awareness of the risk of hepatitis viral infections across the globe including Ghana [51], most of the street beauticians and barbers change razor blades on each customer during nail or hair care. As far as this practice is a good one, it is still prudent for public health activities to continuously focus on sensitizing customers to demand regular safety services during nail or hair care.

Finally, the study indicated that almost one-third of the participants demonstrated overall safe occupational practices for the prevention of viral hepatitis. This finding suggested better practices than cosmetologists of Gondar town in Ethiopia, where 15% of them were reported to conduct safe work practices [46]. The higher safety practices of participants in this study as compared to Ethiopia can partly be attributed to coronavirus disease 2019 (COVID-19) control measures. At the time of the data collection, it was mandatory by all workers as a social responsibility to protect themselves and the well-being of their customers amid the COVID-19 pandemic. It is also possible that participants modified their practices when they became aware that they were being observed during data collection, though social desirability bias was controlled. Consequently, risk assessment audits need to be carried out across street beauticians' and barbers' working premises so that further information can be gathered to implement well-tailored interventions to improve their practices.

On the other hand, greater non-adherence to safety occupational practices among street beauticians and barbers in our study could be linked to their poor knowledge levels due to an established long-standing correlation between low knowledge levels and poor safety work practices and vice versa [13–15, 52]. Moreover, these unsafe practices can be attributed to the lack of training especially on safety and IPC measures, and the absence of regulatory systems to monitor their work practices in the various Districts/Municipal Assemblies in Ghana which was similarly recommended by the EHOs. Most countries like Italy and Pakistan have well-implemented regulatory policies on safety practices where beauticians and barbers are regulated through structured training, licensing, and monitoring systems [21, 43]. Some African countries like Tanzania and Egypt have occupational and public health requirements for all beauticians and barbers [11, 49], however, there is no existence of such policies or requirements for these street cosmetologists in Ghana [23, 53]. It is established that the provision of legislation and safety regulations is necessary for beauticians and barbers to comply with safety

measures, as they provide guidance and security to improve health and safety at work [49]. This calls for the Government of Ghana and its relevant authorities to pay superior attention to the activities of street beauticians and barbers in the fight against viral hepatitis but should not only be fixated on mobilizing taxes from them.

## Factors associated with knowledge level and occupational practices

Quantitatively, while street beauticians' and barbers' knowledge levels about hepatitis infections were determined by education, daily customers attended to, and work experience, their safety work practices were enhanced by occupational type. The present study revealed that participants who had higher education up to the tertiary level were more likely to have good knowledge about viral hepatitis (HBV and HCV) infections. This finding corroborates a study done in Pakistan where higher educational status was found to be associated with greater hepatitis awareness among cosmetologists [16]. Higher education does not only expose individuals to a wider range of information on infectious diseases but also prepares them to be effective, efficient, and reliable in preventing infections in the work environment in most communities [26]. Literate persons are more likely to participate in outreach screening and education programs on viral hepatitis and also most health promotional activities are targeted to them due to their perceived comprehension or knowledge of health issues [54].

Beauticians and barbers who had a workload of more than 20 customers per day and had worked for at least four years were more likely to have good knowledge about HBV and HCV which supports the results of a similar study in South Asia [16] where increased workload as well as work experience were correlated with good knowledge level about viral hepatitis. It implies that attending to more customers per day increases workers' performance rate and gives them greater experience which consequently leads to attaining increased practical capacities and adequate knowledge about the prevention of viral hepatitis infections like HBV and HCV [55].

Besides, beauticians were more likely to provide safe occupational services as compared to barbers in the present study. This is an expected finding as most of the beauticians in our study were females and there is evidence that female beauticians effectively provide a safer work environment and services to their customers [44, 56]. There was virtually no strong link between the qualitative arm and the predictors of knowledge or occupational practices among the street beauticians and barbers, and thereby future studies should be conducted to explore this weakness. Notwithstanding, it is imperative for Ghanaian policy targeted at street beauticians and barbers to consider higher education levels in the licensing and regulation of these local cosmetologists.

## Limitations of the study

The study participants' perceived risk of COVID-19 among themselves and mandatory preventive measures put in place at the time of data collection, such as regular hand washing and regular use of disinfectants, may have influenced participants' practices. It is also possible that participants modified their practices when they became aware that they were being observed during data collection, though efforts were put in place to reduce social desirability biases. The snowballing technique as part of the multistage sampling method used in the recruitment of study participants could lead to selection bias. Notwithstanding, the mixed methods nature of our study concurrently gives a joint interpretation of both quantitative and qualitative data which makes it unique.

## Conclusions

The study revealed a high level of awareness of viral hepatitis, however, a larger proportion of the street beauticians and barbers had limited knowledge about HBV and HCV. Tertiary level of education, heavy daily workload, and increased work experience were predictors of good knowledge level while beauticians adhered to safety measures or practices at work. This study found that the overall safety practices of the participants were very poor and posed a great risk for transmission of viral hepatitis through the use of ineffective disinfectants, irregular hand hygiene practices, and improper handling of tools. The Government of Ghana and policy-makers should institute policies and guidelines on the health and safety practices for street beauticians and barbers. The Ministry of Health in collaboration with the MLGRD should organize regular obligatory practical training on safety measures including IPC and regularly conduct workplace inspections for street beauticians and barbers. The relevant regulatory agency (Department of Environmental Health) under the MLGRD should also be provided with the necessary resources to monitor the activities of street beauticians and barbers in the communities.

## Supporting information

**S1 Dataset. Quantitative dataset.**
(XLSX)

**S2 Dataset. Qualitative dataset.**
(MP3)

## Acknowledgments

The authors are grateful to all research assistants, respondents, and volunteers who greatly contributed to this study. We are also thankful to the Volta Regional Health Directorate, Ghana Health Service–Ghana, for their unwavering support during data collection.

## Author Contributions

**Conceptualization:** Silas Adjei-Gyamfi, Abigail Asirifi, Clotilda Asobuno, Felix Kwame Korang.

**Data curation:** Silas Adjei-Gyamfi, Abigail Asirifi, Clotilda Asobuno, Felix Kwame Korang.

**Formal analysis:** Silas Adjei-Gyamfi, Clotilda Asobuno, Felix Kwame Korang.

**Funding acquisition:** Felix Kwame Korang.

**Investigation:** Silas Adjei-Gyamfi, Abigail Asirifi, Clotilda Asobuno, Felix Kwame Korang.

**Methodology:** Silas Adjei-Gyamfi, Abigail Asirifi, Clotilda Asobuno, Felix Kwame Korang.

**Project administration:** Silas Adjei-Gyamfi, Felix Kwame Korang.

**Resources:** Silas Adjei-Gyamfi, Felix Kwame Korang.

**Software:** Silas Adjei-Gyamfi, Felix Kwame Korang.

**Supervision:** Silas Adjei-Gyamfi, Abigail Asirifi, Clotilda Asobuno, Felix Kwame Korang.

**Validation:** Silas Adjei-Gyamfi, Abigail Asirifi, Clotilda Asobuno, Felix Kwame Korang.

**Visualization:** Silas Adjei-Gyamfi, Abigail Asirifi, Clotilda Asobuno.

**Writing – original draft:** Silas Adjei-Gyamfi, Abigail Asirifi, Felix Kwame Korang.

**Writing – review & editing:** Silas Adjei-Gyamfi, Abigail Asirifi, Clotilda Asobuno, Felix
Kwame Korang.

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
