## [Decision Letter · Decision Letter 0]

29 Jul 2024

PONE-D-24-25615Knowledge and occupational practices of beauticians and barbers in the transmission of viral hepatitis: a mixed-methods study in Volta Region of GhanaPLOS ONE

Dear Dr. Adjei-Gyamfi,

Thank you for submitting your manuscript to PLOS ONE. After careful consideration, we feel that it has merit but does not fully meet PLOS ONE’s publication criteria as it currently stands. Therefore, we invite you to submit a revised version of the manuscript that addresses the points raised during the review process.

**The reviewers have submitted their comments on your manuscript. Based on their recommendations, major revisions are required. Please revise your manuscript accordingly and address all the reviewers' comments **

We look forward to receiving your revised manuscript.

Kind regards,

Phuping Sucharitakul

Academic Editor

PLOS ONE

Journal Requirements:

This study was supported by The Project for Human Resource Development Scholarship, Japan International Cooperation Agency (JICA), and Nagasaki University School of Tropical Medicine and Global Health, Japan.

3. We are unable to open your Supporting Information file "PONE_S1 quan data.dta". Please kindly revise as necessary and re-upload.

**Additional Editor Comments:**

Dear Authors,

The reviewers have submitted their comments on your manuscript. Based on their recommendations, major revisions are required. Please revise your manuscript accordingly and address all the reviewers' comments before we can consider it for publication.

Reviewers' comments:

Reviewer's Responses to Questions

**Comments to the Author**

1. Is the manuscript technically sound, and do the data support the conclusions?

Reviewer #1: Yes

Reviewer #2: Partly

2. Has the statistical analysis been performed appropriately and rigorously? 

Reviewer #1: No

Reviewer #2: No

3. Have the authors made all data underlying the findings in their manuscript fully available?

Reviewer #1: Yes

Reviewer #2: No

4. Is the manuscript presented in an intelligible fashion and written in standard English?

Reviewer #1: Yes

Reviewer #2: No

5. Review Comments to the Author

**Reviewer #1: **Reviewer’s comments

Manuscript ID: PONE-D-24-25615

Title: Knowledge and occupational practices of beauticians and barbers in the transmission of viral hepatitis: a mixed-methods study in Volta Region of Ghana

General comments

Given the enormity of the impact of viral hepatitis on quality of life, morbidity and mortality, this topic remains relevant, and therefore, I wish to commend that authors for a good job done. The authors are however advised to meticulously proofread their work and address a few specific comments.

Specific comments

Abstract

Background

1. The opening sentence should be revised as it appears not to articulate the problem. The authors can consider stating the problem as “Hepatitis B and C viral infections are endemic in Ghana. Also, the National Policy on Viral Hepatitis stipulates that there is unreliable data, limited knowledge, and deficiency in research on viral hepatitis, especially among high risk workers in the eastern part of the country”

Methods

2. Please break this sentence into two to read well: “While an in-depth interview was used to collect data from five environmental health officers who were selected as key informants in the qualitative stage, structured questionnaires/checklists, and direct observations were employed to collect data from 340 beauticians and barbers in the quantitative stage”.

Conclusion

3. The conclusion sub-section of the abstract is good except that I recommend that, the authors add a brief sentence on recommendations regarding how the problem can be addressed.

Introduction

4. Lines 116-117: Please provide a citation to the statement “The ability to prevent HBV and HBC infections among beauticians and barbers as well as their customers is highly dependent on having adequate knowledge about these infections.”

5. Lines 128-129: Please provide relevant citation to “…however, other practical routes of infection including the operations of beauticians and barbers could be a source of transmission”

Methods

Study area

6. The authors have aptly given a brief profile of the study area. However, I recommend that the authors write on the healthcare delivery systems in the study area, including specialized arrangements for hepatitis B and C testing if available.

Sample size determination

7. The authors estimated the sample size of the quantitative arm of the study using a design effect size = 1.5. Can the authors justify why such design effect size was chosen? What was the guiding principle?

8. The authors did not mention how the sample size for the qualitative arm of the study was obtained. Authors are advised to address it.

Data collection and variables

9. How were the questionnaire, interview guide and checklist designed? What was the content of each instrument? Was the questionnaire validated and tested for reliability? The authors are strongly advised to elaborate on the data collection instruments.

10. The authors failed to explicitly state the variables of the study. For the purposes of clarity, authors are advised to clarify what variables were being used in their study.

Statistical analysis

11. Lines 245-247: What guided the use of median as cut-off point? Did the authors visualize the data? Was the data normally distributed or not. Please elaborate your choice of median as the measure of central tendency. As a guide, use mean as measure of central tendency for normally distributed data and median for skewed data.

12. Referring to Tables 4 & 5, it is obvious that some cells of the contingent tables have fewer counts. Consequently, biased estimates have been obtained as large odds ratios and wide confidence intervals. I strongly advise that the authors account for such sparse data which have produced large odds ratios and wide confidence intervals. Authors may address this concern by referring to Balegha (2024). https://doi.org/10.1371/journal.pgph.0003160; Heinz (2010). https://doi.org/10.1002/sim.3794 and Firth (1993). https://doi.org/10.1093/biomet/80.1.27

Discussion

13. The authors attempted a great discussion of the study findings. However, the authors have not adequately espoused and discussed similar and contrasting findings of previous studies. Therefore, the authors could not elaborate on the practical and policy implications of the results. I therefore recommend that the authors thoroughly discuss their findings taking note of similar findings, contrasting findings, explanation for such discrepancies, the practical import of the findings, theoretical and policy aspects of the results if any.

**Reviewer #2:** Topic: Knowledge and Occupational Practices of Beauticians and Barbers in the Transmission of Viral Hepatitis: A Mixed-Methods Study in the Volta Region of Ghana

Manuscript Number: PONE-D-24-25615

Reviewer: ZB

Detailed Comments

Background

Lines 145-147: The original text states: “Furthermore, there is unreliable data, limited knowledge, and a deficiency in research with no established occupational safety regulations on viral hepatitis prevention and control, especially in the Volta region of Ghana (1).” This suggests that the study is focused on policy and regulation issues. Please revise this to align with the study's intention, which is to investigate knowledge and practices regarding viral hepatitis.

Methods

Sampling:

• Statement: “To recruit the 340 beauticians and barbers, five districts’ capital towns were selected using simple random sampling at the initial stage.”

Comment: The explanation of how simple random sampling was used to recruit respondents is unclear. Please specify the sampling unit.

• Population Clarification: Clarify the term “the population size of each selected town.” Does this refer to the resident population or the population targeted by the study?

• Snowball Sampling: The use of a snowball sampling procedure from an “available member list” to select participants needs clarification. Why was this technique used instead of obtaining a list of beauticians and barbers, which are identifiable business activities? How was the sample distributed between beauticians and barbers?

Qualitative Component:

• Sample Size: Explain how the sample size for the qualitative component was determined. Include details about the qualifications and training of data collectors. Also, clarify the duration of each interview, as 20 minutes seems too short and potentially unproductive.

Observation:

• Sample Determination: How was the observation sample determined? Were observations conducted over different days of the week? Address how social desirability bias was managed, considering that respondents were aware of being observed.

Analysis:

• Knowledge and Practice Scores: The method for estimating absolute knowledge and practice scores using a median cut-off point is confusing. How do Cronbach’s alpha coefficients of 0.81 and 0.79 relate to establishing a cut-off point for knowledge scores? Please revisit and clarify this.

Qualitative Analysis:

• Coding and Triangulation: Clarify whether the analysis was manual or software-assisted. Explain how the code structure was defined and established, the number of coders involved, and how agreement was achieved among them. Additionally, explain how the results were triangulated with the quantitative findings.

Data Collection Tools/Instrument:

• Comprehensiveness: Include comprehensive information about the data collection tools, including validity (content-wise), the number of items, measurement methods and scoring techniques, and the operational definitions of knowledge and practices.

Results

• Table 1: The qualitative background should not be included in this table and should be presented separately.

• Presentation of Findings: The results from the quantitative and qualitative analyses are presented separately, with no integration of the findings. This section should be revised to integrate and triangulate both results effectively.

Discussion

• Integration Required: The discussion currently separates findings from the quantitative and qualitative analyses. This approach is incorrect. The discussion should be integrated, analyzing and discussing both sets of findings together to draw meaningful conclusions and implications. A careful revision is required in this section.

6. PLOS authors have the option to publish the peer review history of their article (what does this mean?). If published, this will include your full peer review and any attached files.

Reviewer #1: No

Reviewer #2: No

---

## [Author Response · Author response to Decision Letter 0]

18 Sep 2024

The Editor-in-chief,

Plos One.

RESPONSES TO JOURNAL REQUIREMENTS AND REVIEWERS’ COMMENTS

Many thanks for kindly reviewing our manuscript. We have carefully considered all concerns raised and incorporated feedback, where possible. The quality of the revised manuscript has greatly improved as a result of your useful comments. We have provided point-by-point responses to the respected observations and comments raised as exhibited below. Please note that the changes in the revised manuscript have been highlighted in red. 

Thank you.

Yours Sincerely,

Silas Adjei-Gyamfi

(Corresponding author)

JOURNAL REQUIREMENTS

[Comment 1] When submitting your revision, we need you to address these additional requirements. Please ensure that your manuscript meets PLOS ONE's style requirements, including those for file naming. The PLOS ONE style templates can be found at 

[Response 1] Thanks for the comment. The manuscript has been revised to meet PLOS ONE’s style.

[Comment 2] Thank you for stating in your Funding Statement: This study was supported by The Project for Human Resource Development Scholarship, Japan International Cooperation Agency (JICA), and Nagasaki University School of Tropical Medicine and Global Health, Japan. Please provide an amended statement that declares all the funding or sources of support (whether external or internal to your organization) received during this study, as detailed online in our guide for authors at http://journals.plos.org/plosone/s/submit-now. Please also include the statement “There was no additional external funding received for this study.” in your updated Funding Statement. Please include your amended Funding Statement within your cover letter. We will change the online submission form on your behalf.

[Response 2] Thanks for the useful comments. The funding statement has been modified as “This study was funded by The Project for Human Resource Development Scholarship under the Japan International Cooperation Agency (JICA), and Nagasaki University School of Tropical Medicine and Global Health, Japan through MPH scholarship support which was received by FKK from 2019 to 2021. There was no additional external funding received for this study. The funders had no role in study design, data collection and analysis, decision to publish, or preparation of the manuscript”

[Comment 3] We are unable to open your Supporting Information file "PONE_S1 quan data.dta". Please kindly revise as necessary and re-upload.

[Response 3] Thanks for this comment. The supporting file “PONE_S1 quan data.dta” is the dataset for the quantitative arm of the mixed-method study which was collected and analysed for the study. The file has been revised and re-uploaded.

[Comment 4] Your ethics statement should only appear in the Methods section of your manuscript. If your ethics statement is written in any section besides the Methods, please delete it from any other section. 

[Response 4] Thanks for the suggestion. The Ethics statement can only be found in the methods section (see page 13; lines 262–268; ethical considerations; materials and methods)

Reviewer #1

[General comment] Given the enormity of the impact of viral hepatitis on quality of life, morbidity and mortality, this topic remains relevant, and therefore, I wish to commend that authors for a good job done. The authors are however advised to meticulously proofread their work and address a few specific comments.

[General Response] We appreciate the reviewer’s deep comprehension and recognition of the significance of our manuscript. Proofreading has been done now. We are pleased to address the useful comments and suggestions made by Reviewer #1 below.

Abstract

Background

[Comment 1] The opening sentence should be revised as it appears not to articulate the problem. The authors can consider stating the problem as “Hepatitis B and C viral infections are endemic in Ghana. Also, the National Policy on Viral Hepatitis stipulates that there is unreliable data, limited knowledge, and deficiency in research on viral hepatitis, especially among high risk workers in the eastern part of the country”

[Response 1] Thanks for this important suggestion. Having supported your suggestion, the background of the abstract has been revised (see page 2; lines 23–25; abstract; background). 

Methods

[Comment 2] Please break this sentence into two to read well: “While an in-depth interview was used to collect data from five environmental health officers who were selected as key informants in the qualitative stage, structured questionnaires/checklists, and direct observations were employed to collect data from 340 beauticians and barbers in the quantitative stage”.

[Response 2] Thanks for the useful suggestion. The sentence has been divided into two (see page 2; lines 32–35; abstract; methods).

Conclusion

[Comment 3] The conclusion sub-section of the abstract is good except that I recommend that, the authors add a brief sentence on recommendations regarding how the problem can be addressed.

[Response 3] Thanks for the comments. Having supported your suggestion, a brief statement on recommendations has been added (see page 3; lines 55–57; conclusion; abstract). 

Introduction

[Comment 4] Lines 116-117: Please provide a citation to the statement “The ability to prevent HBV and HBC infections among beauticians and barbers as well as their customers is highly dependent on having adequate knowledge about these infections.”

[Response 4] Thanks for the comment. The above sentence has been cited (see page 6; lines 113–114; introduction). 

[Comment 5] Lines 128-129: Please provide relevant citation to “…however, other practical routes of infection including the operations of beauticians and barbers could be a source of transmission”

[Response 5] Thanks for the comment. The above sentence has been referenced as suggested (see page 6; lines 125–126; introduction). 

Methods

Study area

[Comment 6] The authors have aptly given a brief profile of the study area. However, I recommend that the authors write on the healthcare delivery systems in the study area, including specialized arrangements for hepatitis B and C testing if available.

[Response 6] Thanks for the suggestion. Healthcare delivery systems in the study area among others have been added (see pages 8–9; lines 160–172; methods and materials; study area). 

Sample size determination

[Comment 7] The authors estimated the sample size of the quantitative arm of the study using a design effect size = 1.5. Can the authors justify why such design effect size was chosen? What was the guiding principle?

[Response 7] Thanks for the comment. The design effect used in this study was based on the WHO conservative design effect (DEFF) size of 1.5. This DEFF indicates how well a sample of street beauticians and barbers in the region represents the larger population which is employed in most hepatitis related studies. With the level of expected practices and absolute precision including 95% confidence intervals, a conservative DEFF of 1.5 was used to account for any expected increase in variance due to clustering (see page 10; line 193; methods and materials; sample size and sampling). 

[Comment 8] The authors did not mention how the sample size for the qualitative arm of the study was obtained. Authors are advised to address it.

[Response 8] Thanks for the comment. Having supported your comment, qualitative studies do not usually have calculations for sample size but mostly quote the number of participants for sample size. Either a specific number of participants are interviewed or the interviews are done based on saturation during qualitative research. Hence, the qualitative arm of this study used five key informants (one from each study site). Details are explained in the manuscript (kindly see page 10; lines 213–218; methods and materials; sample size and sampling).

Data collection and variables

[Comment 9] How were the questionnaire, interview guide and checklist designed? What was the content of each instrument? Was the questionnaire validated and tested for reliability? The authors are strongly advised to elaborate on the data collection instruments.

[Response 9] Thanks for this useful comment. Having supported your suggestion, the data collection section has been revised in the manuscript (see page 12; lines 232–251; methods and materials; data collection and variables)

[Comment 10] The authors failed to explicitly state the variables of the study. For the purposes of clarity, authors are advised to clarify what variables were being used in their study.

[Response 10] Thanks for this comment. A table has been added to describe the key quantitative variables (see Table 2; page 15).

Statistical analysis

[Comment 11] Lines 245-247: What guided the use of median as cut-off point? Did the authors visualize the data? Was the data normally distributed or not. Please elaborate your choice of median as the measure of central tendency. As a guide, use mean as measure of central tendency for normally distributed data and median for skewed data.

[Response 11] Thanks for this in-depth suggestion. The knowledge and practice scores of participants were first visualized to check their distribution of which both were skewed (non-normal) distribution. Hence, the median was appropriate for the analysis. Precise elaboration is shown in the manuscript now (see page 16; lines 308–310; methods and materials; statistical analysis).

[Comment 12] Referring to Tables 4 & 5, it is obvious that some cells of the contingent tables have fewer counts. Consequently, biased estimates have been obtained as large odds ratios and wide confidence intervals. I strongly advise that the authors account for such sparse data which have produced large odds ratios and wide confidence intervals. Authors may address this concern by referring to Balegha (2024). https://doi.org/10.1371/journal.pgph.0003160; Heinz (2010). https://doi.org/10.1002/sim.3794 and Firth (1993). https://doi.org/10.1093/biomet/80.1.27

[Response 12] Thanks for this insightful comment. The statistical analysis section has been revised to address biased estimates (see pages 16–17; lines 317–333; methods and materials; statistical analysis). Moreover, refer to Tables 7 and 8 on pages 30 and 31 respectively.

Discussion

[Comment 13] The authors attempted a great discussion of the study findings. However, the authors have not adequately espoused and discussed similar and contrasting findings of previous studies. Therefore, the authors could not elaborate on the practical and policy implications of the results. I therefore recommend that the authors thoroughly discuss their findings taking note of similar findings, contrasting findings, explanation for such discrepancies, the practical import of the findings, theoretical and policy aspects of the results if any.

[Response 13] Thanks for this remarkable comment. The discussion section has been carefully revised to meet the suggestions above (Kindly see pages 32–38; lines 572–705; discussion).

Reviewer #2

Background

[Comment 1] Lines 145-147: The original text states: “Furthermore, there is unreliable data, limited knowledge, and a deficiency in research with no established occupational safety regulations on viral hepatitis prevention and control, especially in the Volta region of Ghana (1).” This suggests that the study is focused on policy and regulation issues. Please revise this to align with the study's intention, which is to investigate knowledge and practices regarding viral hepatitis.

[Response 1] Thanks for the comments. The above sentence has been revised as “Besides the fact that there is unreliable data and deficient occupational safety regulations on viral hepatitis, there is also limited research on the awareness of hepatitis and work-related practices among street beauticians and barbers, especially in the Volta region of Ghana.” (see page 7; lines 142–145; introduction). 

Methods

Sampling:

[Comment 2] Statement: “To recruit the 340 beauticians and barbers, five districts’ capital towns were selected using simple random sampling at the initial stage.”

Comment: The explanation of how simple random sampling was used to recruit respondents is unclear. Please specify the sampling unit.

[Response 2] Thanks for this insightful comment. Details on the selection procedure has been added to the sampling session (see page 10; lines 197–200; materials and methods; sample size and sampling).

[Comment 3] Population Clarification: Clarify the term “the population size of each selected town.” Does this refer to the resident population or the population targeted by the study?

[Response 3] Thanks for the comment. This is referring to the resident population since there is no available member list in the region. Revisions have been made now (see page 10; lines 201–203; materials and methods; sample size and sampling). Also, see Table 1 on page 11.

[Comment 4] Snowball Sampling: The use of a snowball sampling procedure from an “available member list” to select participants needs clarification. Why was this technique used instead of obtaining a list of beauticians and barbers, which are identifiable business activities? How was the sample distributed between beauticians and barbers?

[Response 4] Thanks for this comment. Snowballing sampling was used for the study because it is very difficult to locate street barbers and beauticians as they do not operate from a fixed place and there was no available member list. This technique was also useful as most of the street barbers and beauticians do not have Bussiness Operating Permit (BOT) and are evading tax payments. Therefore, referral by somebody within their social network increased their trust in participating in the study (see page 10; lines 205–210; materials and methods; sample size and sampling).

Qualitative Component:

[Comment 5] Sample Size: Explain how the sample size for the qualitative component was determined. Include details about the qualifications and training of data collectors. Also, clarify the duration of each interview, as 20 minutes seems too short and potentially unproductive.

[Response 5] Thanks for the beautiful comment. The manuscript has been revised to take care of the above comments. On how sample size was selected, kindly see (page 11; lines 213–218; materials and methods; sample size and sampling). For qualifications and training of data collectors, kindly see (page 12; lines 246–251; materials and methods; data collection and variables). Finally, for the duration of the interview, please see (page 14; lines 285–286; materials and methods; data collection and variables). The time excludes the preparatory and the administrative processes.

Observation:

[Comment 6] Sample Determination: How was the observation sample determined? Were observations conducted over different days of the week? Address how social desirability bias was managed, considering that respondents were aware of being observed.

[Response 6] Thanks for the comment. The observation sample was determined as 340 as shown in the sample size calculation. Observations were conducted over different days of the week. Finally, we preempted the possibility of social desirability bias and took several measures to reduce or mitigate its tendencies throughout the data collection process (see page 13; lines 257–268; materials and methods; data collection and variables).

Analysis:

[Comment 7] Knowledge and Practice Scores: The method for estimating absolute knowledge and practice scores using a median cut-off point is confusing. How do Cronbach’s alpha coefficients of 0.81 and 0.79 relate to establishing a cut-off point for knowledge scores? Please revisit and clarify this.

[Response 7] Thanks for this significant suggestion. Having supported your suggestion, clarifications have been made. The knowledge and practice scores of participants were first visualized to check their distribution of which both were 

---

## [Decision Letter · Decision Letter 1]

20 Oct 2024

PONE-D-24-25615R1Knowledge and occupational practices of beauticians and barbers in the transmission of viral hepatitis: a mixed-methods study in Volta Region of GhanaPLOS ONE

Dear Dr. Adjei-Gyamfi,

Thank you for submitting your manuscript to PLOS ONE. After careful consideration, we feel that it has merit but does not fully meet PLOS ONE’s publication criteria as it currently stands. Therefore, we invite you to submit a revised version of the manuscript that addresses the points raised during the review process.

**Dear Authors,**Reviewers have provided their comments. Please revise the manuscript accordingly and provide a detailed outline of the changes made in response to each of the reviewers' comments.Regards, 

We look forward to receiving your revised manuscript.

Kind regards,

Phuping Sucharitakul

Academic Editor

PLOS ONE

Journal Requirements:

Reviewers' comments:

Reviewer's Responses to Questions

**Comments to the Author**

1. If the authors have adequately addressed your comments raised in a previous round of review and you feel that this manuscript is now acceptable for publication, you may indicate that here to bypass the “Comments to the Author” section, enter your conflict of interest statement in the “Confidential to Editor” section, and submit your "Accept" recommendation.

Reviewer #1: (No Response)

Reviewer #3: (No Response)

2. Is the manuscript technically sound, and do the data support the conclusions?

Reviewer #1: Yes

Reviewer #3: Yes

3. Has the statistical analysis been performed appropriately and rigorously? 

Reviewer #1: Yes

Reviewer #3: Yes

4. Have the authors made all data underlying the findings in their manuscript fully available?

Reviewer #1: Yes

Reviewer #3: Yes

5. Is the manuscript presented in an intelligible fashion and written in standard English?

Reviewer #1: Yes

Reviewer #3: Yes

6. Review Comments to the Author

Reviewer #1: Reviewer 1 comment on revised manuscript

Full title: Knowledge and occupational practices of beauticians and barbers in the transmission of viral hepatitis: a mixed-methods study in Volta Region of Ghana

Manuscript ID: PONE-D-24-25615R1

Comment

I wish to congratulate the authors for a great job done in addressing the comments raised in the original manuscript.

However, the authors are encouraged to address comment 8 of the previously mentioned- on the guiding principle for the number of study participants selected in the qualitative arm of the study.

As rightly pointed out by the authors there is no specific formula for sample size determination in qualitative research except for the principle of data saturation. I think that the authors should state in the manuscript that data saturation was used as the guiding principle for recruitment.

Once again congratulations!!!

Reviewer #3: Manuscript is interesting with an important but usually neglected cause of blood borne diseases. I recommend this manuscript theme and topic be published to increase the accessibility and expansion with regards to this topic in the future.

The quantitative data is presented, but in a different available format, thus i am unable to review or comment with regards to the quantitative data used for the manuscript.

The qualitative data is accessible via MP3.

Manuscript is written in fairly good English, with only some correction and final proofreading required.

Comments to improve the manuscript:

- Author should define "beautician" and "barber" and explore why these 2 different profession were selected as the main subject of the study, and are they or are they not considered "high-risk workers"

- Author should explain why a mixed method and triangulation analysis was used in this study, and how the qualitative data can add more impact to the findings.

-Author should define the category of findings such as cut-offs for high vs low awareness, good vs poor knowledge

-Author should also include the how the initial beautician/barber was chosen from the premise, especially if there was more than one

-Suggest make the keywords one word, and singular and include Ghana to make it more easily and wider reach and accessibility

7. PLOS authors have the option to publish the peer review history of their article (what does this mean?). If published, this will include your full peer review and any attached files.

Reviewer #1: **Yes: **Augustine Ngmenemandel Balegha

Reviewer #3: No

---

## [Author Response · Author response to Decision Letter 1]

23 Oct 2024

Dear Editor-in-chief,

RESPONSE TO REVIEWERS’ COMMENTS

Many thanks for the second and rigorous review of our manuscript. We have carefully addressed all concerns raised and provided feedback, where possible. Due to your significant comments, the quality of the revised manuscript has been enormously enhanced. We have therefore provided point-by-point responses to the dignified comments raised as shown below and also highlighted all revised points in the manuscript in red.

Yours Sincerely,

Silas Adjei-Gyamfi

(Corresponding author)

JOURNAL REQUIREMENT

Comment: Please review your reference list to ensure that it is complete and correct. If you have cited papers that have been retracted, please include the rationale for doing so in the manuscript text, or remove these references and replace them with relevant current references. Any changes to the reference list should be mentioned in the rebuttal letter that accompanies your revised manuscript. If you need to cite a retracted article, indicate the article’s retracted status in the References list and also include a citation and full reference for the retraction notice.

Response: Thanks for this vital comment. The reference list is complete and correct. Our manuscript is free from any retracted papers or articles.

REVIEWER #1

General comments: I wish to congratulate the authors for a great job done in addressing the comments raised in the original manuscript. However, the authors are encouraged to address comment 8 of the previously mentioned- on the guiding principle for the number of study participants selected in the qualitative arm of the study. As rightly pointed out by the authors there is no specific formula for sample size determination in qualitative research except for the principle of data saturation. I think that the authors should state in the manuscript that data saturation was used as the guiding principle for recruitment. Once again congratulations!!!

Response: Many thanks for acknowledging the revisions made and commending the good efforts of the authors. We also appreciate your significant comments and deep comprehension of our manuscript which have greatly contributed to the improvement of our manuscript. As you suggested, a sentence has been added to take care of the principle of data saturation used in the qualitative arm of the study (see page 11; lines 221–222; methods and materials; sample size and sampling).

REVIEWER #3

General comments: Manuscript is interesting with an important but usually neglected cause of blood borne diseases. I recommend this manuscript theme and topic be published to increase the accessibility and expansion with regards to this topic in the future. The quantitative data is presented, but in a different available format, thus i am unable to review or comment with regards to the quantitative data used for the manuscript. The qualitative data is accessible via MP3. Manuscript is written in fairly good English, with only some correction and final proofreading required.

Response: Many thanks for reviewing our revised manuscript. We are grateful for your significant comments and deep comprehension of our manuscript which have greatly contributed to the improvement of our manuscript. The quantitative data is in Microsoft Excel. Furthermore, two different English-proficient persons have gone through the revised manuscript and have addressed the majority of the writing issues.

Comment 1: Author should define "beautician" and "barber" and explore why these 2 different profession were selected as the main subject of the study, and are they or are they not considered "high-risk workers"

Response 1: Thanks for this noble comment. Beauticians and barbers and their jobs have been described and/or defined in the manuscript (kindly see page 5; lines 97–103; introduction). In the present era especially in Ghana, the jobs of beauticians and barbers are interrelated. In the past years, beauticians only styled the hair of women but now they shave and style hair and also perform foot and nail care for all kinds of gender. Additionally, barbers who were known to only shave the hair of men are now shaving, styling, and sometimes providing nail care services to their customers irrespective of their gender. Geographically, people across the globe call them different names such as “beauticians”, “barbers”, “hairdressers”, “hairstylers” or “nail caretakers”. Ghanaians interchangeably label them as either “beauticians” or “barbers”. Meanwhile, the majority of them have no formal training in cosmetology and/or podiatry. This accounts for the reason why the authors sometimes label them as “local cosmetologists” in the manuscript. 

Beauticians and barbers are at higher risk for blood-borne infections like hepatitis (especially B and C) because they provide services to a larger population without knowing the health status of these populations. These workers are religiously prone to occupational hazards such as being cut by used sharp tools like razor blades, and knives while or after attending to their customers. It therefore makes it very interesting to study about them and their job practices in the transmission of hepatitis infections. Compared to other street workers in Ghana like shoe repairers, food vendors/hawkers, and general goods hawkers among others, beauticians and barbers are at higher risk. Moreover, they are less recognized and minor groups where much attention has not been placed on them. Most of the focus and hepatitis-related studies are placed on maternal and child health (pregnant women, nursing mothers, neonates, etc) and commercial sex workers among other cohorts. Meanwhile, these workers are mostly ignored. On the other hand, it became necessary to find out whether these workers are trained on safety measures and whether their occupational practices are regulated or monitored. 

Comment 2: Author should explain why a mixed method and triangulation analysis was used in this study, and how the qualitative data can add more impact to the findings.

Response 2: Thanks for the comment. As explained in the study design (kindly see page 9; lines 175–185), a mixed-method approach is vital in this study. According to Koopmans (2016), mixed methods is the preferable approach for addressing complex issues and exploring the linkage of factors operating at the individual, community, institutional, and societal levels. In the context of this study, little is known about the study, and using mixed methods was the best approach to describe the situation and provide answers to the study objectives. (Koopmans, M., 2016. Mixed Methods in Search of a Problem, Journal of Mixed Methods Research; Vol. 11, Issue 1, pp. 16 – 18. DOI: 10.1177/1558689816676662). The quantitative method is very necessary to provide proven statistical data whereas the qualitative study offers the opportunity to provide meaning behind the numbers. On the other hand, the study was carried out concurrently. Thus, a parallel study of both quantitative and qualitative strands within the same study. Triangulation analysis of the survey, observations, and key Informant Interviews was done to form a joint interpretation of the data. The qualitative arm gave more explanation on why the unsafe occupational practices are predominant among the study participants as well as the poor knowledge level exhibited by these street workers. For example, the qualitative arm revealed that there is a lack of licensing, monitoring, and training for street beauticians and barbers. Hence the poor practices and knowledge are attributed to the lack of regulatory systems and training on safety measures for these street beauticians and barbers as indicated by the qualitative arm. 

Comment 3: Author should define the category of findings such as cut-offs for high vs low awareness, good vs poor knowledge

Response 3: Thanks for the vital comment. Good or poor knowledge was determined by using the median cut-off point of 14 knowledge-related items as described in Table 2. This means that a knowledge-related score of < 7 was regarded as “poor knowledge” while a score of ≥ 7 was regarded as “good knowledge” (see Table 2; page 15).

As seen in Table 5 (page 25), majority of the study participants (88.2%) were aware or had ever heard of hepatitis virus infections. This indicates that most of the participants had a high awareness level about hepatitis infections. Awareness in this study is used as “realizing or knowing that something (e.g. Hepatitis infection) exists” according to Oxford Learner’s Dictionary (kindly see; https://www.oxfordlearnersdictionaries.com/definition/english/awareness). It can also mean “consciousness”, “mindfulness”, “attention”, “alertness” as explained by Merriam-Webster dictionary (https://www.merriam-webster.com/dictionary/awareness). Hence, the awareness in this study was not categorized into low or high but rather based on frequency proportions. 

Comment 4: Author should also include the how the initial beautician/barber was chosen from the premise, especially if there was more than one

Response 4: Thanks for the significant suggestion. The first street barber or beautician who met the inclusion criteria was recruited by approaching him/her while providing services to a customer in the community (see page 10; lines 208–209; methods and materials; sample size and sampling).

Comment 5: Suggest make the keywords one word, and singular and include Ghana to make it more easily and wider reach and accessibility

Response 5: Thank you for the comment. Keywords have been made singular as much as possible and Ghana is already included as a keyword (kindly see page 4; lines 70–76: Keywords).

---

## [Editor Report · Decision Letter 2]

28 Oct 2024

Knowledge and occupational practices of beauticians and barbers in the transmission of viral hepatitis: a mixed-methods study in Volta Region of Ghana

PONE-D-24-25615R2

Dear Dr. Adjei-Gyamfi,

We’re pleased to inform you that your manuscript has been judged scientifically suitable for publication and will be formally accepted for publication once it meets all outstanding technical requirements.

Kind regards,

Phuping Sucharitakul

Academic Editor

PLOS ONE
---

## [Editor Report · Acceptance letter]

31 Oct 2024

PONE-D-24-25615R2 

PLOS ONE

Dear Dr. Adjei-Gyamfi, 

I'm pleased to inform you that your manuscript has been deemed suitable for publication in PLOS ONE. Congratulations! Your manuscript is now being handed over to our production team.

Kind regards, 

on behalf of

Dr. Phuping Sucharitakul 

Academic Editor

PLOS ONE